# Antibiotic Therapy for Difficult-to-Treat Infections in Lung Transplant Recipients: A Practical Approach

**DOI:** 10.3390/antibiotics11050612

**Published:** 2022-05-02

**Authors:** Lorena van den Bogaart, Oriol Manuel

**Affiliations:** Infectious Diseases Service and Transplantation Center, Lausanne University Hospital and University of Lausanne, 1011 Lausanne, Switzerland; lorena.van-den-bogaart@chuv.ch

**Keywords:** MDR *Pseudomonas aeruginosa*, *Burkholderia cepacia* complex, *Mycobacterium abscessus* complex, nocardiosis, lung transplant

## Abstract

Lung transplant recipients are at higher risk to develop infectious diseases due to multi-drug resistant pathogens, which often chronically colonize the respiratory tract before transplantation. The emergence of these difficult-to-treat infections is a therapeutic challenge, and it may represent a contraindication to lung transplantation. New antibiotic options are currently available, but data on their efficacy and safety in the transplant population are limited, and clinical evidence for choosing the most appropriate antibiotic therapy is often lacking. In this review, we provide a summary of the best evidence available in terms of choice of antibiotic and duration of therapy for MDR/XDR *P. aeruginosa*, *Burkholderia cepacia* complex, *Mycobacterium abscessus* complex and *Nocardia* spp. infections in lung transplant candidates and recipients.

## 1. Introduction

Despite improvements in immunosuppressive regimens and antimicrobial preventive strategies, infectious complications remain a significant cause of morbidity and mortality after lung transplantation [1]. Bacterial infections may also play a role in the development of chronic lung allograft dysfunction (CLAD), a heterogeneous range of syndromes that lead to allograft loss in lung transplant recipients, affecting up to 40% of patients within five years [2]. Moreover, lung transplant recipients are at higher risk for developing difficult-to-treat bacterial infections than transplant recipients from other organs. Lung transplant candidates are often chronically colonized or infected before transplantation with pathogens such as *P. aeruginosa*, *Burkholderia cepacia* complex (BCC) and non-tuberculous mycobacteria (NTM). These pathogens are difficult to eradicate, leading to a higher risk of re-colonization and poorer outcome after transplantation. Antibiotic therapy for these difficult-to-treat infections may be challenging due to reduced efficacy for multidrug resistant (MDR) pathogens, drug-drug interactions between antibiotics and immunosuppressive agents, and comorbidities that result in higher rates of antibiotic-associated toxicity.

Multidrug resistance is a worldwide growing problem; in 2020, in Europe, up to 17% of *P. aeruginosa* isolates were resistant to at least two or more antimicrobial groups with some countries reporting carbapenems resistance rates of more than 50% [3]. MDR is defined as resistance to at least one agent in three antibiotic classes; extensive drug resistance (XDR) as non-susceptibility to at least one agent in all but one or two antibiotic classes; and pan-drug resistance (PDR) to all agents in all classes [4]. The incidence of infections produced by MDR/XDR *P. aeruginosa* strains is higher in transplant recipients than in the general population [5,6]. In a cohort study involving 503 patients, 18% of *P. aeruginosa* isolates in bloodstream infections were MDR in non-transplant recipients as compared to 43% among transplant recipients, i.e., a 3.47-fold higher risk of being infected by an MDR strain [7].

The management of infections caused by MDR/XDR organisms is a major challenge for clinicians in real life. Despite the development of several novel antibiotics, the armamentarium against MDR/XDR pathogens remains limited and clinical evidence for guiding the choice of the best therapeutic options is scarce. In selecting the antibiotic regimen, clinicians should not only consider the pathogen’s resistance pattern but also additional elements, such as drug tolerability (in particular for infections requiring prolonged therapy such as nocardiosis or MABc infections), pharmacokinetics, and drug-drug interactions.

In this narrative review, we summarize the existent literature and we provide a clinical practical approach on the best antibiotic therapy for selected difficult-to-treat infections whose management is particularly challenging in lung transplant recipients, namely MDR/XDR *P. aeruginosa* (in particular *P. aeruginosa* with “difficult to treat resistance˝, see definition in Section 2), *Burkholderia cepacia* complex, *Mycobacterium abscessus* complex and *Nocardia* spp.

## 2. MDR/XDR *Pseudomonas aeruginosa* Infection

The prevalence of infections caused by MDR *P. aeruginosa* among transplant recipients varies according to the geographical area and the type of transplant performed [5]. In the Swiss Transplant Cohort Study, a national cohort of transplant recipients, *P. aeruginosa* was responsible for 9% of bacterial infections (23% of whom were MDR) during the first year post-transplant. A Spanish study showed that up to 63% (31/49) of *P. aeruginosa* bloodstream infections were XDR [8]. Colonization of the respiratory tract by MDR *P. aeruginosa* is common in the pre-transplant period in candidates to lung transplantation with cystic fibrosis (CF), having a high risk of subsequent post-transplant infection. This was demonstrated in a study where 75–88% of patients were re-colonized with the same strain of *P. aeruginosa* within a median time of 23 days after transplantation [9].

### 2.1. Mechanism of Resistance in Pseudomonas aeruginosa

The increasing prevalence of MDR/XDR *P. aeruginosa* is due to the extraordinary ability to develop resistance predominantly mediated by chromosomal mutations, resulting in hyperproduction of inducible chromosomal cephalosporinase AmpC, loss or reduction of the OprD porin and overexpression of efflux pumps [10]. Horizontal acquisition of resistance determinants such as carbapenemase or *ESBL* genes is less common, but it is increasingly observed. Recent studies showed that most carbapenemase- or ESBL-producing strains belong to the so-called high-risk clones, mainly ST235, ST111 or ST175 [11]. The spread of metallo-beta-lactamases (MBL) strains (specifically VIM and IM) is particularly concerning due to resistance to ceftazidime/avibactam and ceftolozane/tazobactam (see below). Moreover, several of these mutations often co-exist in MDR/XDR *P. aeruginosa*. Figure 1 summarizes the mechanisms of antibiotic resistance in *P. aeruginosa*. In addition to the classic antimicrobial susceptibility tests assessed by disk diffusion or e-test, determining the molecular mechanism of carbapenem resistance may help to select the most suited antibiotic therapy. In the case of a phenotypic carbapenem resistant strain, further assessment of the resistance pattern by biochemical and molecular assays is recommended [12], even if most of these tests are limited to detection of beta-lactamases, and not of porin loss or efflux pumps hyperexpression.

### 2.2. Respiratory Tract Colonization by P. aeruginosa in Lung Transplant Recipients: To Eradicate or Not?

The relationship between chronic colonization and/or infection by *P. aeruginosa* and the development of CLAD, including bronchiolitis obliterans syndrome (BOS) and restrictive allograft syndrome (RAS) as a clinical phenotype, remains controversial. Several studies have shown that persistent allograft colonization with *P. aeruginosa* (in particular de novo colonization after transplantation) is associated with development of BOS and reduced graft survival [13,14,15,16]. Therefore, determining whether antibiotic treatment of colonization with *P. aeruginosa* in lung transplant recipients is associated with a better outcome is of importance. Currently, the majority of data associating *P. aeruginosa* eradication with a slowing of the lung function decline comes from non-transplanted patients with CF rather than from lung transplant recipients. In chronically infected CF patients with *P. aeruginosa*, two Cochrane systematic reviews suggest a potential benefit of inhaled antibiotic treatment for early and chronic *P. aeruginosa* infection in terms of microbiological eradication from the respiratory tract (2 randomized placebo-controlled trials) or improvement of lung function compared to placebo (in 4 of the 11 randomized placebo-controlled trials). However, the overall quality of evidence was limited because of large differences in terms of study design, outcomes, and sample size [17,18]. Moreover, there is insufficient evidence to determine whether the antibiotic eradication strategy had an impact on other outcomes, such as quality of life, adverse events, and overall survival.

Several different antibiotics are generally used in this context. The Cystic Fibrosis Foundation strongly recommends inhaled tobramycin for 28 days for the treatment of initial or new growth of *P. aeruginosa* from the respiratory tract [19]. In a recent meta-analysis, aztreonam lysine in combination with tobramycin inhalation solution was associated with improved changes in FEV1 and sputum density in CF patients [20]. Finally, two randomized clinical trials showed similar efficacy between aerosolized colistin and tobramycin [18,21]. Among lung transplant recipients, randomized clinical trials evaluating the effect of eradication treatment are lacking. A large retrospective study showed better CLAD-free and graft survival in lung transplant recipients with successful eradication treatment (after first *P. aeruginosa* isolation) compared to unsuccessful eradication [22]. In other studies, *P. aeruginosa* re-colonization after lung transplantation had no impact on post-transplant morbidity and mortality [9,23], and aerosolized anti-pseudomonas therapy was not associated with increased *Pseudomonas*-free survival [24,25].

Despite the lack of robust evidence in transplant population and due to this potential benefit, lung transplant programs generally recommend therapy with aerosolized antibiotics in the post-transplant setting to prevent or treat colonization by *P. aeruginosa*. The use of nebulized colistin is common in the immediate post-transplant period if *P. aeruginosa* is isolated from respiratory samples in order to protect the bronchial suture. In addition to a favorable pharmacokinetics profile, inhaled antibiotics have the advantage of a reduced nephrotoxicity in the context of calcineurin inhibitor use [5].

### 2.3. Antibiotic Therapy for MDR/XDR P. aeruginosa Infection

Management of infections caused by MDR/XDR *P. aeruginosa* is a major therapeutic challenge which often requires infectious disease consultation. Both MDR/XDR *P. aeruginosa* infections and lung transplant recipients are largely underrepresented in the majority of clinical trials assessing the efficacy of new antimicrobial agents. Most of the data in the transplant population comes from case reports and case series.

Empirical combination therapy with agents from different antimicrobial classes is generally recommended to increase the probability to administer at least one active agent. Once the beta-lactam agent demonstrates in vitro activity, combination therapy seems not to be superior to monotherapy [26,27,28]. Therefore, combination therapy is not recommended for lung transplant recipients with MDR/XDR *P. aeruginosa* that are still susceptible to a beta-lactam agent [28]. The maintenance of a second agent, such as aminoglycosides or colistin, increases the likelihood of antibiotic-associated adverse events. In case of prior colonization of the donor or recipient, empirical therapy should be based on the susceptibility profile of *P. aeruginosa* isolates before transplantation [5]. When beta-lactams are used, high-dose and prolonged-/continuous-infusion administration is recommended [27]. Table 1 shows the suggested antibiotic regimens according to mechanism of resistance.

The Infectious Disease Society of America (IDSA) and the European Society of Clinical Microbiology and Infectious Diseases (ESCMID) have defined the category of “difficult to treat resistance˝ (DTR) in their guidelines for the treatment of MDR Gram negative infections [28,29]. According to these guidelines, DTR *P. aeruginosa* are those strains non-susceptible to piperacillin/tazobactam, ceftazidime, cefepime, aztreonam, meropenem, imipenem, ciprofloxacin, and levofloxacin. In case of infection caused by DTR isolates, ceftolozane/tazobactam, ceftazidime/avibactam, and imipenem/relebactam as monotherapy are recommended by IDSA guidelines as first-line therapy. ESCMID guidelines recommend only ceftolozane/tazobactam as the antibiotic of choice, because of insufficient evidence for the use of imipenem/relebactam and ceftazidime/avibactam. Clinical outcomes from trials directly comparing the effectiveness of these three agents are not available, but given that the rate of *P. aeruginosa* isolates susceptible to ceftolozane/tazobactam is generally higher than for the other antibiotics, ceftolozane/tazobactam can be considered as the preferred option in this setting. Ceftolozane/tazobactam therapy was associated with a 20% improvement in clinical cure and a 28% decrease in acute kidney injury compared with a polymyxin or aminoglycoside-based regimen for the treatment of resistant *P. aeruginosa* infections [30]. Moreover, clinical cure rates in a cohort of immunocompromised patients treated with ceftolozane/tazobactam for MDR *P. aeruginosa* infections were similar to those reported in non-immunocompromised patients [31]. Relebactam, a new beta-lactamase inhibitor given in combination with imipenem, restored susceptibility to imipenem in 70% of imipenem-non-susceptible isolates of *P. aeruginosa* due to ampC hyperproduction, efflux overexpression, and porin OprD loss [32]. Clinical data of imipenem/relebactam in lung transplant recipients is lacking.

In addition to the three previously mentioned antibiotics, cefiderocol, a novel siderophore cephalosporin, is an alternative option, but it should be reserved as a second-line choice in case of clinical or microbiological failure [28,33]. Cefiderocol overcomes several resistance mechanisms, including enzymatic hydrolysis by serine- and metallo-carbapenemases, porin channel mutation and efflux pump overproduction. A clinical trial comparing cefiderocol with the best available therapy for the treatment of carbapenem-resistant Gram–negative infections (including *P. aeruginosa* in 24% of cases) showed similar clinical and microbiological efficacy, but higher mortality in the cefiderocol group, in particular for *A. baumanii* infection [34]. If DTR *P. aeruginosa* strain is resistant to ceftolozane/tazobactam (as in the case of ST-235 clone with expression of GES enzyme, a class A beta-lactamase), continuous infusion of ceftazidime/avibactam or cefiderocol in monotherapy is recommended. Preclinical data support a possible synergism with fosfomycin, although this needs to be confirmed in clinical trials [33,35].

MBL expression by *P. aeruginosa* represents one of the most challenging resistance patterns that clinicians have to manage. In the case of MBL isolates, treatment options are based only on expert opinion and case reports. MBL-producing *P. aeruginosa* led to lung lobectomy and death in two lung transplant recipients [36,37]. Possible antibiotic strategies for these infections include monotherapy with cefiderocol or colistin, combination therapy with cefiderocol plus inhaled colistin (in case of pneumonia), and a combination of colistin, fosfomycin and aminoglycosides [10,33]. Unlike *Enterobacteriacea*, aztreonam/avibactam seems not to be active against MBL-producing *P. aeruginosa* [38]. However, in vitro data showed synergistic effects and restoration of bactericidal activity against MBL-producing *P. aeruginosa* nonsusceptible to both aztreonam and ceftazidime/avibactam when combining aztreonam with ceftazidime/avibactam [39].

ESCMID guidelines suggest treatment with two in vitro active drugs when polymyxins, aminoglycosides, or fosfomycin are used for DTR *P. aeruginosa* treatment [29]. There is controversial clinical evidence on the benefit of nebulized colistin for the treatment of MDR gram-negative ventilator-associated pneumonia [40]. While two meta-analyses suggested that the use of nebulized and IV colistin was associated with better clinical and microbiological outcomes than intravenous therapy alone [41,42], another meta-analysis published in 2018 did not confirm these findings [43].

Finally, although there are little clinical data on the impact on transplant outcomes of inappropriate antibiotic prophylaxis at the time of transplant, it is recommended that surgical prophylaxis should be adapted to the susceptibility profile in case of colonization of the donor respiratory track by *P. aeruginosa*.

## 3. *Burkholderia cepacia* Complex (BCC) Infection

BCC includes distinct species, previously called genomovars. Although these species are phylogenetically and phenotypically indistinguishable, they widely differ in virulence [44]. *B. cenocepacia* and *B. multivorans* account for approximately 85% of BCC infection in many countries. *B. cenocepacia* is the species associated with the highest antibiotic resistance and mortality rates, with a one-year survival rate in lung transplant recipients ranging from 37% to 60% [45,46]. Outcomes of recipients infected with other non-*B. cenocepacia* species, including *B. multivorans*, are similar to those of uninfected patients [46,47,48]. Limited evidence suggests *B. gladioli* (which is not a member of BCC) and *B. dolosa* may be associated with worsened survival post-lung transplantation. A retrospective study reported a survival rate in patients colonized with *B. dolosa* at one-, three- and five-year post-transplant of 73%, 53% and 30%, respectively, a lower survival compared to that of the CF population without BCC, but higher compared to survival of patients colonized with *B. cenocepacia* [49].

Colonization with BCC is considered a contraindication to lung transplantation in many centers because of the risk of “cepacia syndrome”, a syndrome of necrotizing pneumonia with sepsis associated with extremely poor outcomes [50]. A meta-analysis including 11 studies showed that BCC colonization in the pre-transplant period was the most robust variable associated with increased early post-transplant mortality [51]. Due to the extensive antibiotic resistance profile, therapeutic options are limited and no guidelines exist. A careful evaluation of BCC species and susceptibility testing in referral centers is mandatory before denying access to lung transplantation.

### 3.1. Antibiotic Therapy for Burkholderia cepacia Complex (BCC) Infection

Efflux pumps, reduced outer membrane permeability and production of beta-lactamases are the most frequent mechanisms of resistance in BCC. Prolonged multidrug regimens are usually prescribed depending on susceptibility tests and local experience. Table 2 reports the main antibiotic options for BBC infection. A study analyzing 50 *B. multivorans* strains in vitro and in vivo showed that 70% were MDR and 22% XDR. Greater than 90% of the isolates were resistant to tobramycin, imipenem, and ciprofloxacin and more than 30% to the two ‘first-line’ agents for the treatment of BCC infections, ceftazidime and trimethoprim–sulfamethoxazole (TMP/SMX). On the contrary, all strains were susceptible to ceftazidime/avibactam [52]. *B. cenocepacia* has even higher rates of resistance, with 86% of isolates being MDR [53]. Avibactam is a potent inhibitor of the class A carbapenemases, PenA, one of the major resistance determinants expressed in *B. multivorans* [52]. The experience in real-life on the use of ceftazidime/avibactam for the treatment of BCC infections in lung transplant recipients is currently limited to case reports, showing encouraging results [54,55,56].

In the case of resistance to ceftazidime/avibactam, the combination of piperacilline/avibactam (using piperacilline/tazobactam plus ceftazidime/avibactam) may overcome the mechanism of resistance (avibactam inhibits PenA and piperacilline inhibits ampC beta-lactamases) and restore the susceptibility to ceftazidime and piperacilline, as demonstrated in vitro [57]. Imipenem/relabactam is also active against 70% of the ceftazidime/avibactam-resistant BCC [58]. Cefiderocol represents an additional therapeutic option with nearly 95% of all isolates of BCC demonstrating in vitro susceptibility [59]. Because clinical data supporting in vivo efficacy are still lacking, cefiderocol may be used in the absence of alternative antibiotic options. Finally, temocillin is characterized by a potent activity against *Burkholderia* spp. because it is poorly hydrolyzed by endogenous beta-lactamases, resulting in activity against up to 87% of MDR *Burkholderia* strains [60].

### 3.2. Other Therapeutic Approaches

Irrigation of the chest cavity and bronchi at the time of transplant with taurolidine was associated with a reduced colonization by MDR pathogens (including BCC) at one-year post-transplant, but without improvement in allograft function and cumulative survival [61]. Nebulized taurolidine did not have any clinical benefit in BCC colonized patients in a randomized, double-blinded placebo-controlled trial [62]. Finally, bacteriophage treatment in addition to antibiotic therapy may be an additional strategy for treating BCC infections (see Section 4).

## 4. *Mycobacterum abscessus* Complex (MABc) Infection

*Mycobacterium abscessus* complex (MABc) represents an emerging cause of pulmonary and disseminated infection in lung transplant recipients with CF and chronic lung disease. In a large French cohort including more than 1500 CF patients, 3% and 1.45% of patients were colonized by MABc and by *M. avium* complex (MAC), respectively causing pulmonary disease in 80% of cases [63]. Lung transplant recipients are at higher risk of NTM disease compared to other organ transplant recipients, MABc being the most common NTM during the first three years after lung transplantation [64].

### 4.1. Microbiology and Resistance Mechanism

MABc belongs to the group of rapidly growing mycobacteria and includes three subspecies: *M. a. abscessus*, *M. a. massiliense* and *M. a. bolletii*. *M. a. abscessus* is the most common among the subspecies in Europe and the United States [63,65]. Subspecies differentiation is essential because of therapeutic implications, in particular for the different susceptibility pattern to macrolides. Macrolide resistance in MABc can develop through chromosomal mutations in the 23S rDNA (rrl) gene resulting in a high level of resistance or, more frequently, through induction of the *erm(41)* gene, which causes inducible resistance in the presence of a macrolide. The majority of *M. a. abscessus* and *bolletii* strains express an active *erm(41)* gene, which leads to macrolide resistance despite initial in vitro susceptibility. On the contrary, *M. a. massiliense* is characterized by a nonfunctional *erm(41)* gene and macrolide susceptibility. This explains the poorer outcome in patients infected with *M. a. abscessus* compared to *M. a. massiliense* in terms of symptomatic, radiological and microbiological response [66]. In two studies including 99 and 145 patients, the initial sputum conversion ranged from 25 to 31% and from 50 to 88% in patients infected with *M. a. abscessus* and *massiliense*, respectively [66,67]. Recurrence occurred in 6/11 patients (54%) with *M. a. abscessus* lung disease who achieved initial culture conversion, as compared to 18% of patients with *M. a. massiliense* [68]. Historically, pulmonary infection with MABc was considered in many transplant centers as a contraindication to lung transplantation due to poor post-transplant outcomes. This depended also on the isolated species; *M. a. abscessus* was associated with a worse outcome compared to the other species, as shown in a study where five out of seven patients colonized with *M. a. abscessus* died after lung transplantation [69]. However, thanks to some case-series reporting successful outcomes, this practice has been questioned by many experts [70]. A study including 13 lung transplant candidates colonized with MABc showed an overall survival of 77% one year after transplantation [71]. Currently, very few transplant centers consider infection with MABc as an absolute contraindication for listing; however, 76% regard it as a relative contraindication [72].

### 4.2. Antibiotic Therapy for MABc Infection in Lung Transplant Recipients

Eradication of MABc infection in chronically infected patients may be attempted before lung transplantation. However, because the majority of patients remain culture-positive at the time of transplantation due to the urgency of transplantation and/or to failure of clearing the infection, the continuation of antibiotic treatment is necessary in the post-transplant period as well. Transient colonization without clinical disease usually does not require antibiotic treatment and is not associated with impaired allograft outcomes [73].

The optimal therapeutic regimen for MABc disease has not been evaluated in clinical trials. Current guidelines are mostly based on case series and expert opinion; treatment in lung transplant recipients seems not to differ from the non-transplant population. Generally, a prolonged therapy composed by a more aggressive initial phase of combined parenteral and oral drugs for four to eight weeks when bacterial burdens are greater, followed by a maintenance phase of at least two oral drugs and an inhaled antibiotic for at least 12–18 months is recommended [74].

Treatment regimens differ based on susceptibility to macrolide, as shown in Figure 2. IDSA guidelines suggest an initial regimen consisting of ≥three active drugs in macrolide susceptible disease with at least one intravenous agent and one oral agent in addition to azithromycin. In case of macrolide resistance, a ≥four-drug regimen in initial phase consisting of a combination of amikacin plus one or two additional parenteral agents and two or three oral agents is recommended [74].

Intravenous amikacin is the most effective parenteral agent against MABc, and it is included in all regimens with a dosing of three times per week, with close monitoring of toxicity. The other parenteral options include imipenem and tigecycline. A study demonstrated in vitro enhanced activity against MABc if carbapenems were associated with vaborbactam or relebactam, with a significant reduction in MIC values. Therefore, meropenem-vaborbactam and imipenem-relebactam are expected to substitute the use of a carbapenem alone in the treatment of MABc [75].

Oral agents with in vitro activity against MABc, but limited clinical experience, include linezolid (300 to 600 mg daily), tedizolid (200 mg daily), clofazimine (100 mg daily), and bedaquiline (400 mg daily for two weeks followed by 200 mg three times per week). Clofazimine was associated with a favorable outcome in a series of 12 lung transplant recipients with MABc pulmonary disease, with a good tolerability profile [73]. There is no evidence that fluoroquinolones (i.e., moxifloxacin) are efficacious against any of *M. abscessus* subspecies, so that it is currently not recommended to be included in the initial therapeutic regimens [64]. Use of linezolid is often limited by side effects, such as cytopenia and peripheral neuropathy. Whereas tedizolid seems to be an alternative to linezolid, a recent study evaluating prolonged exposure to tedizolid in transplant recipients found no safety benefit [76]. Omadacycline, a new oral tetracycline, has shown in vitro and in vivo efficacy against MABc, with a 75% clinical success observed in a series of 12 patients treated with omadacycline [77].

Inhaled liposomal amikacin is widely used and it was associated with clinical improvement in terms of early and sustained negative sputum cultures in MAC and MABc lung disease in a randomized controlled trial [78].

### 4.3. Other Treatments

The benefit of surgical therapy is controversial with no significant differences in cure rate across different studies, including MAC lung disease. Indication of surgery may be evaluated in some specific cases, such as sputum conversion failure, sputum relapse after initial conversion and complications like recurrent hemoptysis [79].

Lytic bacteriophage therapy represents a promising strategy against MDR/XDR infections in lung transplant candidates and recipients. Recent case reports have described the successful use of adjunctive bacteriophage therapy for treatment of MDR *P. aeruginosa*, *B. dolosa*, *M. abscessus* and *Achromobacter xylosoxidans* infections in six lung transplant candidates and recipients [80]. A three-phage anti-*M. abscessus* cocktail was administered in a lung transplant recipient with disseminated *M. abscessus* infection with a favorable microbiological and clinical response, without adverse events [81]. Clinical trials for phage therapy in CF patients are currently underway (NCT04684641).

## 5. Nocardiosis

Nocardiosis is a rare, life-threating opportunistic infection affecting 0.04% to 3.5% of patients after solid-organ transplantation in Europe [82]. The highest rates among transplant recipients is observed after lung transplantation, probably due to the higher net state of immunosuppression and direct contact of the environment with the allograft. In a multicenter European cohort study of transplant recipients, nocardiosis occurred after a median of 17.5 months, more than 80% of patients presented with lung disease, and patients with disseminated nocardiosis had mainly central nervous system and mucocoutaneous involvement. Because around 40% of cases with cerebral involvement are asymptomatic, systematic cerebral imaging is mandatory in all cases of nocardiosis [83].

### Therapeutic Management

Nocardiosis in transplant recipients is difficult to manage, because of a lack of high quality data on the best therapeutic options in this population, and because it requires a long-term therapy that is usually associated with significant toxicity and drug-drug interactions. Microbiological identification of *Nocardia* spp. and a susceptibility test may be difficult to obtain in real life, thus requiring clinicians to choose an empirical treatment based on infection severity and local epidemiology (Table 3).

In the case of primary skin or non-severe lung disease, several authors recommend monotherapy [84,85], generally with TMP/SMX. A closed monitoring of renal function is mandatory in particular in transplant recipients receiving calcineurin-inhibitors because of potential additional nephrotoxicity. Some species of *Nocardia* are resistant to TMP/SMX (*N. farcinica*, *N. otitidiscaviarum*), thus identification at the species level and antimicrobial susceptibility testing are strongly recommended to guide therapy. However, a susceptibility test of TMP/SMX should be interpreted with caution because of limited data correlating proposed breakpoints with clinical outcomes. In case of resistance for sulfonamide at broth microdilution, additional methods to confirm resistance are required [84]. Linezolid monotherapy represents another interesting option being active in vitro against all *Nocardia* species.

Combination therapy with at least two agents is recommended in the case of severe disease and/or CNS involvement to ensure at least one agent is effective. Generally, the initial multidrug regimen includes imipenem, amikacin, and TMP/SMX [85]. Imipenem is more active than either meropenem or ertapenem in mouse models. The combination with amikacin seems to be more effective in the treatment of cerebral and pulmonary nocardiosis than TMP-SMX alone. Third-generation cephalosporines as empirical treatment are not recommended due to the high resistance rates of some species, like *N. farcinica* [84]. Because of its broad activity against various *Nocardia* species and the excellent CNS penetration, linezolid may be included in an initial combination therapy in severe infection. A good tolerability and safety profile was observed in patients treated with linezolid for a median duration of 28 days [86]. The initial treatment should be administered intravenously for at least three to six weeks and/or until clinical improvement is documented. A switch to an oral maintenance regimen with TMP/SMX, minocycline or amoxicillin-clavulanate based on susceptibility test is then indicated for a duration of 6–12 months.

While low dose TMP/SMX given as a prophylaxis to *Pneumocystis jirovecii*, infection does not prevent nocardiosis [82], it has not been correlated with the emergence of TMP/SMX-resistant strains in transplant recipients [87].

## 6. Expert Opinion and Conclusions

Lung transplant recipients are often colonized or infected with difficult-to-treat pathogens, which complicates the management of these infections in routine clinical practice. New antibiotic options are currently available, but data among transplant recipients are limited, so the optimal antibiotic strategy in this population is often poorly defined. Moreover, careful use of new antibiotics in selected patients by means of specific stewardship programs is important to avoid the development of further antibacterial resistance.

In case of pre-operative chronic airway colonization with *P. aeruginosa*, antibiotic prophylaxis during surgery should be selected according to the most recent susceptibility tests. While universal prophylaxis with inhaled colistin in the early post-transplant period is used in some transplant centers, we do not administer it routinely due to lack of evidence. We suggest using inhaled colistin for three to six months only to treat de novo colonization with *P. aeruginosa*. In the case of de novo *P. aeruginosa* infection, an empirical combination antibiotic regimen based on local epidemiology and risk factor for MDR is necessary. In countries with high rates of carbapenem resistance, a rapid diagnostic test detecting carbapenemase-producer pathogens may be useful in a peri-transplant setting in order to rapidly adapt the empiric antibiotic therapy. In case of DTR *P. aeruginosa* infection, in addition to phenotypical susceptibility test, we recommend to perform biochemical or molecular assays able to identify the resistance mechanisms in order to select the best antibiotic treatment. Ceftolozane/tazobactam monotherapy according to a susceptibility test is the first line regimen for DTR *P. aeruginosa* infection.

Lung transplant candidates colonized by BCC must be carefully evaluated for transplantation; in our center, this is not an absolute contraindication to transplant. The decision to list the patient for lung transplantation should consider the BCC species as well as its resistance patterns. Susceptibility tests should include new antibiotic options, namely ceftazidime/avibactam and imipenem/relebactam. We suggest a combination therapy with ceftazidime/avibactam, imipenem/relebactam and TMP/SMX in case of *B. cenocepacia* infection. A concomitant sinus surgery that is often the cause of rapid lung re-colonization is recommended as well.

The management of MABc infection should first include the subspecies identification, followed by an assessment of macrolide susceptibility. The latter requires an extended incubation of isolates in the presence of clarithromycin to detect the expression of the *erm(41)* gene. Molecular assays can also reveal a mutation in the 23S rRNA genes that is associated with high-level resistance to macrolides. Treatment of MABc infection requires a combination regimen of parenteral and oral antibiotics, often associated with a poor tolerability and non-negligible toxicity. New alternative schedules of therapy with potential better tolerability (i.e., replacing tigecycline with omadacycline) or with improved efficacy (i.e., adding relebactam to imipenem) are currently available, but more data in real life are required.

A promising strategy for the treatment of MDR/XDR infection is phagotherapy. Although successful outcomes have been reported in some case reports and series, several steps are still required before the use of phagotherapy in routine clinical practice. This includes the production of phages according to good manufacturing practices, the implementation of phages banks for the selection of the most compatible phage or phage cocktail, and the inclusion of patients in well-designed clinical trials. We suggest in selected cases with a failure of antibiotic treatment that referring the patient to a specialized center for evaluation of phagotherapy be considered.

Finally, a multidisciplinary team including pneumologists, thoracic surgeons, and infectious disease consultants is strongly recommended in order to optimize the management of lung transplant recipients with difficult-to-treat infections.

## Figures and Tables

**Figure 1 antibiotics-11-00612-f001:**
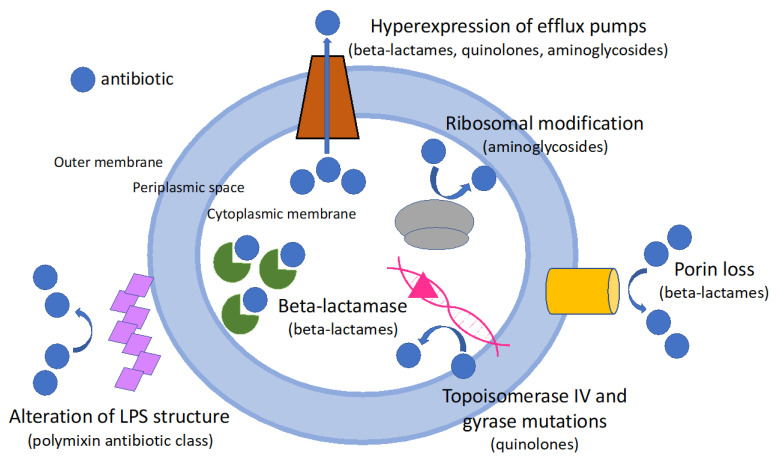
Mechanisms of antibiotic resistance of *P. aeruginosa*. LPS: lipopolysaccharide.

**Figure 2 antibiotics-11-00612-f002:**
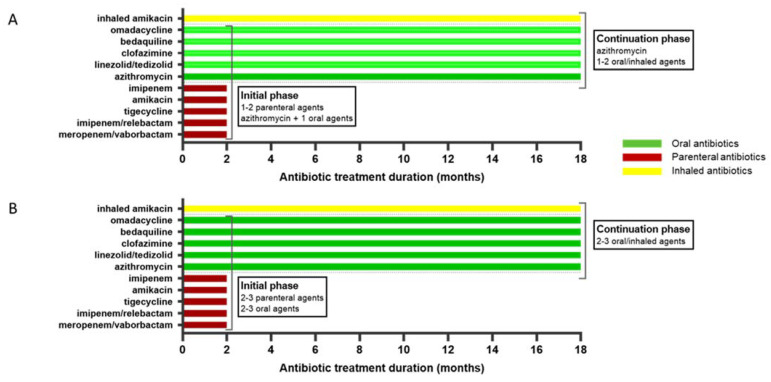
Antibiotic schedule for *Mycobacterium abscessus* complex infection. (**A**) Macrolide resistant strain. (**B**) Macrolide susceptible strain.

**Table 1 antibiotics-11-00612-t001:** Antibiotic treatment options for *P. aeruginosa* infection outside of the urinary tract.

	First-Line Treatment	Other Options
ESBL *P. aeruginosa*	Meropenem 1–2 g q8h (3 h-infusion)	Ceftolozane/tazobactam 1.5 g q8h (for infection other than pneumonia); 3 g q8h (for pneumonia) Ceftazidime/avibactam 2.5 g q8h Imipenem/relebactam 1 g q6h
DTR *P. aeruginosa* (not MBL-producer)	Ceftolozane/tazobactam 3 g q8h (3 h-infusion) Ceftazidime/avibactam 2.5 g/qh (3 h-infusion) Imipenem/relebactam 1.25 g q6h (30 min-infusion)	Cefiderocol 2 g q8h (3 h-infusion)
DTR *P. aeruginosa* (not MBL-producer; resistant to ceftolozane/tazobactam)	Ceftazidime/avibactam 2.5 g q8h (3 h-infusion)	Cefiderocol 2 g q8h (3 h-infusion) Ceftazidime/avibactam 2.5 g q8h (3 h-infusion) + Fosfomycin 12–24 g per day
DTR *P. aeruginosa* (MBL-producer) *	Cefiderocol 2 g q8h (3 h-infusion) Colistin 9 × 10^6^ IU per day Cefiderocol 2 g q8h (3 h-infusion) + inhaled colistin 0.5–2 × 10^6^ q12h	Ceftazidime/avibactam 2.5 g q8h + aztreonam 2 g q8h (3 h-infusion) Colistin + fosfomycin + aminoglycoside Bacteriophage therapy

DTR: difficult-to treat resistance; ESBL: extended spectrum beta-lactamase; MBL: metallo-beta-lactamase. * Optimal treatment is unknown; infectious disease consultation is strongly recommended.

**Table 2 antibiotics-11-00612-t002:** Antibiotic treatment option for *Burkholderia cepacia* complex (BCC) infection.

	First-Line Treatment	Alternative Treatment
BCC	Ceftazidime 2 g q8h TMP/SMX 8–10 mg/kg/day divided q8h or q6h Levofloxacine 750 mg q24h	Minocyclin 100 mg bid Meropenem 2 g q8h
MDR BCC *	Ceftazidime/avibactam 2.5 g q8h	Cefiderocol 2 g q8h
MDR BCC resistant to ceftazidime/avibactam *	Imipenem/relabactam 1.25 g q6h Piperacilline/tazobactam 4.5 g q6h + ceftazidime/avibactam 2.5 g q8h	Cefiderocol 2 g q8h Temocillin 2 g q8h Bacteriophage

BCC: *Burkholderia cepacia* complex, MDR: multi-drug-resistant, TMP/SMX: trimethoprim/sulfamethoxazole. * Combination therapy with at least two to three active molecules is recommended.

**Table 3 antibiotics-11-00612-t003:** Therapeutic management of nocardiosis according to clinical presentation.

Localization	Empiric Induction Treatment *^,±^	Maintenance Oral Therapy ^±^	Duration
Primary skin Pulmonary stable	TMP/SMX orally Linezolid orally	TMP/SMXM inocycline Amoxicillin/clavulanate	6–12 months
Pulmonary moderate/severe	TMP/SMX iv + imipenem OR amikacin TMP/SMX iv + ceftriaxone ± linezolid Linezolid+ ceftriaxone OR imipenem	TMP/SMX Minocycline Amoxicillin/clavulanate	6–12 months
CNS involvement	TMP/SMX iv + imipenem ± amikacin TMP/SMX iv + imipenem + linezolid Linezolid + imipenem Imipenem + amikacin	TMP/SMX	9–12 months
Disseminated (>two organs without CNS involvement)	TMP/SMX iv + imipenem OR amikacin TMP/SMX iv + linezolid + imipenem OR amikacin Imipenem + amikacin	TMP/SMX Minocycline Amoxicillin/clavulanate	6–12 months

TMP/SMX: trimethoprim/sulfamethoxazole; CNS: central nervous system. * Continue multi-drug parenteral therapy for two to six weeks and adjust based on susceptibility test. ^±^ Antibiotic dosing: TMP/SMX 15 mg/kg (divided in three to four doses), linezolid 600 mg q12h, imipenem 500 mg q6h, minocycline 100–300 q12h, amikacin 20–30 mg/kg/day, ceftriaxone 2 g q24h.

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
