# Peer review of "Antibiotic Therapy for Difficult-to-Treat Infections in Lung Transplant Recipients: A Practical Approach"

_antibiotics, 2022, doi:10.3390/antibiotics11050612_

Round 1

Reviewer 1 Report

I will like to congratulate the authors for doing a great job summarizing an extensive albeit important area in lung transplantation. The review is well-written and will be useful for any physician that manages lung transplant patients.

Author Response

We thank the reviewer for the kind comments.

Reviewer 2 Report

Very well-written review on the approach to difficult-to-treat infection after lung transplantation. I have only a few minor comments:

1) I'd suggest to specify also in the title that the review is focused on difficult-to-treat infections

2) While in the P. aeruginosa section the authors correctly introduce readers to the concept of "difficult-to-treat resistance", this concept is already present in the introduction and there could be some confusion in readers regarding the difference between "difficult-to-treat resistance" and "difficult-to-treat", with the latter seeming to include also other considerations such as drug-drug interactions

3) to my understanding (but I am not a microbiologist), piperacillin is not an inhibitor, but rather a walk inducer of AmpC

4) "Despite lung transplant recipients are at high risk of MDR P. aeruginosa infections, they are not included in the majority of clinical trials assessing the efficacy of new antimicrobial agents ". Not only lung transplants recipients are underrepresented in RCTs, but also MDR/XDR infections in general are underrepresented 

5) Please add a legend for colors in the figure

6) Regarding MDR/XDR P. aeruginosa the authors discuss IDSA guidelines. It would be of interest also if the could provide some comments on the recently released ESCMID guidelines for the management of resistant Gram-negative bacteria

7) I agree that rapid molecular tests for resistance determinants are useful within diagnostic algorithms for MDR/XDR P. aeruginosa infections. However, most tests are limited to detection of beta-lactamases, whereas also porin alterations and efflux pumps may play an important role in P. aeruginosa DTR. Readers should be aware of this.

Author Response

We thank the reviewer for the review of our manuscript and for these very welcome suggestions.

1) we thank the reviewer for the great suggestion. We modified the title as suggested.

2) we thank the reviewer for the excellent comment and we agree with him. In order to avoid a misundestanding between the concept of 'difficult-to-treat infection' and 'difficult-to-treat resistance P. aeruginosa' we anticipated in the introduction at line 57 the specific concept of  'difficult-to-treat resistance P. aeruginosa' and in the section 2.3, at line 150, we modified 'difficult-to-treat (DTR) infections' with '“difficult to treat resistance˝(DTR)'. 

3) in the paper published on Journal of Clinical Microbiolog, Zeiser et al. demonstrated in vitro susceptibility to the combination of avibactam and  piperacilline in 13 of 14 Burkholderia isolates that tested nonsusceptible to ceftazidime-avibactam . In order to understand the mechanism of susceptibility of this combination,  they assess the activity of piperacillin and avibactam against PenA1 and AmpC1 beta-lactamases from Burkholderia multivorans ATCC 17616. They found that piperacillin was a substrate for PenA1, but inhibited AmpC. Conversely, avibactam was unable to inhibit AmpC1 but was a potent inhibitor of PenA1.

4) we thanks the reviewer for the suggestions and we modified at section 2.3, line 129 'Despite lung transplant recipients are at high risk of MDR P. aeruginosa infections, they are not included in the majority of clinical trials' with 'Both MDR/XDR P. aeruginosa infections and lung transplant recipients are largely underrepresented in the majority of clinical trials' .

5) We added the legend as suggested.

6) We thanks the reviewer for the great suggestions. We added at section 2.3, line 148, line155 and line 195 several recomandations of ESCMID guidelines.

7) We agree with the reviewer. We added in the section 2.1 line 86 ' further assessment of the resistance pattern by biochemical and molecular assays is recommended [12], even if most of these tests are limited to detection of beta-lactamases, and not of other most frequent alterations as porin loss or efflux pumps hyperexpression.

Reviewer 3 Report

Overall, this is a well-written manuscript. I do not see any major issues except a few minor things. For example, in line 74: ESBL (gene name) should be italic. The manuscript should be read thoroughly and similar issues should be fixed.

Also, there are at least three tables that summarize key information from different contexts but an illustration would greatly improve the overall quality of the manuscript. For example, an illustration depicting the mechanism of resistance in  P. aeruginosa and the proposed therapy based on the current literaure would be a great option to include.

Author Response

We thank the reviewer for the really interesting comments.

We apologized for form errors. We put in italic all the name of genes and species of pathogens.

Concerning the proposed illustration of the mechanism of resistance of P. aeruginosa, we agree with the reviewer. We added a figure with the most important mechanisms of resistances and the resistant antibiotic. We left the proposed treatment according to mechanism of resistance in the table. 

Reviewer 4 Report

The manuscript is written in a concise, clear way, it is well rounded, informative and completely up to date with relevant information supported by appropriate refferences and expertise of the authors.

There some minor formating errors ( i.e.. abstract subheadings should be ommited). I am aware that the journal allows for free format submissions, so now the authors should perform the final formating of the manuscript according to the journal guidelines.

Author Response

We thanks the reviwer for the kind comments. We omitted abstract subheadings  as suggested.